# Research

evolution, theoretical biology,
computational biology

cooperation, altruism, evolution, coevolution,
host–microbiome, mathematical model

**Author for correspondence:**
Lilach Hadany
e-mail: lilach.hadany@gmail.com

# Host–microbiome coevolution can promote cooperation in a rock–paper–scissors dynamics

Ohad Lewin-Epstein and Lilach Hadany

Department of Molecular Biology and Ecology of Plants, Tel-Aviv University, Tel-Aviv 6997801, Israel

OL-E, 0000-0002-8636-9006; LH, 0000-0002-1642-5308

Cooperation is a fundamental behaviour observed in all forms of life. The evolution of cooperation has been widely studied, but almost all theories focused on the cooperating individual and its genes. We suggest a different approach, taking into account the microbes carried by the interacting individuals. Accumulating evidence reveals that microbes can affect their host's well-being and behaviour, yet hosts can evolve mechanisms to resist the manipulations of their microbes. We thus propose that coevolution of microbes with their hosts may favour microbes that induce their host to cooperate. Using computational modelling, we show that microbe-induced cooperation can evolve and be maintained in a wide range of conditions, including when facing hosts' resistance to the microbial effect. We find that host–microbe coevolution leads the population to a rock–paper–scissors dynamics that enables maintenance of cooperation in a polymorphic state. Our results suggest a mechanism for the evolution and maintenance of cooperation that may be relevant to a wide variety of organisms, including cases that are difficult to explain by current theories. This study provides a new perspective on the coevolution of hosts and their microbiome, emphasizing the potential role of microbes in shaping their host's behaviour.

## 1. Introduction

Cooperative behaviour, such that confers a fitness cost to the acting individual, while providing a benefit to its partner, is observed in all levels of organization—from bacteria to communities of multicellulars. As such, the evolution of cooperation by means of natural selection presents a puzzle [1–6].

A major class of models that can explain the evolution of cooperation was introduced by Hamilton [7], who suggested that natural selection may favour a gene that induces cooperative behaviour if directed towards kin, which are likely to carry other copies of the same gene [8–14]. Another class of explanations suggested reciprocity as a reason for cooperation [15], focusing on the benefit to the cooperating individual. When interactions are recurring, conditional cooperative behaviour can evolve, if directed towards individuals that cooperated in previous rounds—even if they are not relatives [2,16–18]. This includes direct reciprocity (A helps B and B helps A), and indirect reciprocity (A helps B and C helps A, based on the reputation of A) [19,20]. Cooperation was further suggested to be favoured as a signal for the quality of the cooperating individual, resulting in increased social status, mating success, etc. [21,22]. More recent work investigated the effect of population structure and viscosity—affecting, among other things, the probability of repeating interactions and interactions among kin—on the evolution of cooperation [23–28].

The vast majority of theories offering an explanation for the evolution of cooperation share a common attribute, focusing on the genes of the cooperating individual, namely on traits that affect the tendency to cooperate and are

transmitted from parent to offspring. Recently, we suggested an alternative explanation, focusing on the *microbes* carried by the interacting individual, that can be transmitted both from parent to offspring and between interacting hosts. We showed that microbes that induce host cooperation can evolve under wide conditions [29]. Here, we combine the two approaches and consider both the host's genes and its microbiome coevolving with respect to cooperation. We use the term 'cooperation' throughout, but note that similar phenomena were referred to also as 'altruism' in the literature.

Almost all organisms carry microbes that can have dramatic effects over their hosts' well-being and behaviour [30–35]. Recent evidence shows that the gut microbiome can affect the brain via the microbiome–gut–brain axis [36–38], potentially affecting brain development, cognitive function, and behaviours such as social interactions and stress [39–46]. In light of this evidence, we raised the hypothesis that microbes can also affect the tendency of their hosts to cooperate. Note that in this work, we use the term microbe in the most general sense, referring to an element that inhabits an organism, can affect its behaviour, and can be transmitted both vertically and horizontally. Our results can be relevant to any element that applies to these characteristics (e.g. plasmids, viruses, multicellular symbionts).

In cases where microbes perform manipulation on their host, a conflict of interest may arise. This could lead to evolution of host resistance by acquiring traits that negate the microbial manipulation [47–50]. In many cases, such resistance may itself incur a cost to the host [51–54], generating complex dynamics of coevolution of the hosts and their microbes.

In addition to the ability to affect host behaviour, microbes have different ways of moving between hosts. Similarly to genes, microbes can be transmitted vertically, namely inherited, from one individual to its offspring [55–59]. Yet, as opposed to genes of multicellular organisms, microbes can also be transmitted horizontally during interactions between hosts [60–63]. In fact, interactions among hosts, such as feeding, grooming, sharing resources, and co-sheltering, involve close proximity between the hosts, and thus serve as a platform for microbial transmission [64–66]. Due to the ability of microbes to transmit horizontally, they can benefit from inducing their host to help another host, that could be inhabited by their transmitted kin—even when the hosts are not related. In that respect, our theory corresponds to the theory of kin selection, where the relevant 'kin' are not the interacting individuals, but rather the microorganisms that inhabit them.

Previous studies examined the evolution of public goods genes that are encoded on mobile genetic elements that can be transmitted vertically and horizontally between bacteria [67–70]. These studies explored the role of transmission and relatedness on the evolution of bacterial cooperation, both empirically and theoretically. Our work is different in several ways. First, we broaden the perspective and claim that in fact, the evolution of cooperation by horizontally transmitted elements is relevant to almost any species, via its microbiome, or any other symbiont. Second, we emphasize the link between social interaction and horizontal transmission, which we show, enables the evolution of cooperation under wide conditions. Last, we study host–microbe coevolution with respect to cooperation, by accounting for both microbial and host alleles.

Here, we study host–microbe coevolution and analyse the conditions that allow the evolution of microbe-induced cooperation in a population of hosts that can evolve resistance to the microbial effect. In our framework, we consider the abilities of the microbes to be transmitted both vertically and horizontally, the costs and benefits of cooperation, and host resistance to the microbial effect. We find that under a wide range of parameters, microbe-induced cooperation facing host resistance generates rock–paper–scissors dynamics in the population, and allows cooperation to be maintained in polymorphism for a long time.

## 2. Results

### (a) Model description

We model a population of asexual haploid hosts, each carrying one type of microbe, $\alpha$ or $\beta$. Hosts interact in pairs, and microbes can affect their hosts' behaviour: microbes of type $\alpha$ increase the tendency of their hosts to cooperate during interaction, while microbes of type $\beta$ do not have any effect over their host's behaviour. In addition, a bi-allelic locus at the host genome determines the susceptibility of the host to the microbial effect. Hosts carrying allele $S$ are susceptible to the microbe's effect, and act cooperatively when carrying microbes of type $\alpha$. Hosts carrying allele $R$ are resistant to the microbial effect and do not act cooperatively regardless of the microbes they carry. This resistance confers a fitness cost of $0 < \delta < 1$ (the case where the resistance cost depends on the microbe type showed similar dynamics, see electronic supplementary material, S1–S3). We thus model a population with four different types of hosts: $\alpha S$, $\alpha R$, $\beta S$, $\beta R$, defined by the combination of microbe type ($\alpha/\beta$) and host allele ($R/S$).

Each generation, hosts pair randomly and interact with a Prisoner's Dilemma payoff (figure 1a). Note that the microbes do not affect the tendency of the hosts to interact. Thus, cooperators and non-cooperators take part in the same number of interactions. During interaction, the fitness of the hosts changes according to their behaviour: a host that behaves cooperatively (type $\alpha S$ only) pays a fitness cost $0 < c < 1$ while its partner receives a benefit $b > c$. A selfish host does not pay the fitness cost, but receives the benefit if its partner is a cooperator. In addition, horizontal transmission of the microbes might occur during interaction (figure 1b). We denote by $T_\alpha$ the probability of microbes of type $\alpha$ being transmitted from one host to the other during interaction, establishing and replacing the resident microbes, and similarly with $T_\beta$. We assume that transmission of one microbe is independent of the other microbe, and when both occur, they occur simultaneously. We note that $T_\alpha$ and $T_\beta$ encompass the probability of completing the entire transmission process: traversing the physical barrier, competing with the native microbial community, and establishing a colony. We assume that during an interaction, a host behaves according to the allele–microbe combination it carried before the interaction. If horizontal transmission occurs, the new microbes establish and start affecting host behaviour right after the current interaction.

We use discrete models, with non-overlapping generations. At the end of each generation, after interactions and horizontal transmissions take place, the hosts reproduce according to their fitness. Both alleles and microbes are vertically transmitted from hosts to their offspring, and the

Proc. R. Soc. B 287: 20192754

(a)

(b)

**Figure 1.** Model illustration. Each individual hosts one type of microbe, either $\alpha$ (inducing cooperation) or $\beta$ (no effect), and one allele, either $S$ (susceptible to the microbial effect) or $R$ (resistant). Thus, only $\alpha S$ hosts are cooperators. Carrying allele $R$ also confers a fitness cost of $\delta$. During interactions $\alpha S$ hosts cooperate: they pay a fitness cost of $0 < c < 1$, and their partners receive a fitness benefit $b > c$. Additionally, horizontal transmission of microbes may occur during interactions, regardless of the alleles that the hosts possess. We denote by $T_\alpha$ the probability of microbes of type $\alpha$ being transmitted to the other host, establishing and replacing the resident microbes, and similarly with $T_\beta$. (a) Fitness matrix showing the fitness of each host, according to its allele, microbe, and interaction partner, when considering one interaction per host per generation. (b) Possible interactions that yield fitness change, microbe transmission, or both. In brackets are the fitness costs for the hosts: $-\delta$ for hosts with allele $R$, and $-c$ for cooperators ($\alpha S$ hosts). Black arrows represent the fitness benefit ($+b$) that cooperators provide to their partners. Coloured arrows (red and green) represent the probability for microbial horizontal transmission during interactions.

offspring host generation replaces the parent generation. We investigate this general model using two approaches. First, in the *Deterministic model* section, we analyse the dynamics of a fully mixed infinite population. Second, in the *Stochastic models* section, we use computational simulations to analyse the dynamics of a finite population, both fully mixed and spatially structured, and account for stochastic effects, imperfect vertical transmission, multiple interactions per generation, and mutations. The additional details of each model are included below in the relevant sections and in the electronic supplementary material.

## (b) Deterministic model

We first consider an infinite, fully mixed population, where each host has an equal probability to interact with any other individual in the population. Every generation, the population is randomly divided into pairs, and each pair interacts once. Host fitness is determined by the interaction payoff and the resistance cost (figure 1). We describe the change in the frequencies of the four host types from one generation to the next using four iterative equations (see electronic supplementary material, equations S1–S4 in S1). By analysing this system of equations, we study the conditions that allow the evolution and maintenance of cooperative behaviour.

We first find that cooperation can evolve (i.e. $\alpha S$ hosts can increase from rarity) only when:

$$\frac{b}{c} > \frac{1 - T_\beta}{T_\alpha} + \frac{T_\beta - T_\alpha}{T_\alpha c} \tag{2.1}$$

(figure 2a,b and electronic supplementary material S1–S3). This result is consistent with [29], where a similar condition determines the evolution of a microbe inducing cooperation in a population of hosts that are all susceptible to the microbe's effect.

Two major factors support the evolution of microbe-induced cooperation. First, the link between interaction and horizontal transmission, which enables the microbes to direct some of their host's resources towards another host that could be inhabited by their transmitted kin. Second, the ability of the microbes to transmit both horizontally and vertically: while horizontal transmission allows the microbes to help their future kin, vertical transmission allows the microbes to enjoy the increased fitness of a host that received help.

Intuitively, the conditions allowing the evolution of cooperation in the presence of host resistance (here) are never wider than the conditions in a susceptible population [29]. We thus continue by assuming (2.1) is satisfied and analysing the additional conditions that allow host susceptibility to the microbes (allele $S$) to increase in frequency. Denoting the proportions of hosts carrying allele $S$ ($\alpha S$ and $\beta S$ hosts), and of hosts carrying both allele $S$ and microbe $\alpha$ ($\alpha S$ hosts) by $x_S$ and $x_{\alpha S}$, respectively, we find that the proportion of allele $S$ increases from one generation to the next if and only if (see analysis in electronic supplementary material, S3):

$$\frac{x_{\alpha S}}{x_S} < \frac{\delta}{c}. \tag{2.2}$$

When $\delta > c$, condition (2.2) is always satisfied and thus the proportion of allele $S$ increases to fixation for any $x_S > 0$. That is, when the cost of cooperation ($c$) is smaller than the cost of host resistance to the microbial effect ($\delta$), cooperation will fixate in the population (area *II* in figure 2a,b). Furthermore, we find that when $\delta < c$, cooperation can evolve and be maintained in polymorphism, and that a polymorphic equilibrium, when it exists, satisfies $x_{\alpha S}/x_S = \delta/c$. The polymorphism in the host alleles is maintained by a balance between the disadvantage of resistance (paid by hosts carrying allele $R$) and the disadvantage of cooperation (paid by those $S$ hosts that carry microbe $\alpha$).

When $\delta < c$, $\alpha S$ hosts bear an inherent disadvantage—they pay a fitness cost of $c$, while the rest of the hosts pay lower costs of either $\delta$ ($\alpha R$ and $\beta R$ hosts), or no cost at all ($\beta S$ hosts). Yet, we find that a polymorphic equilibrium exists

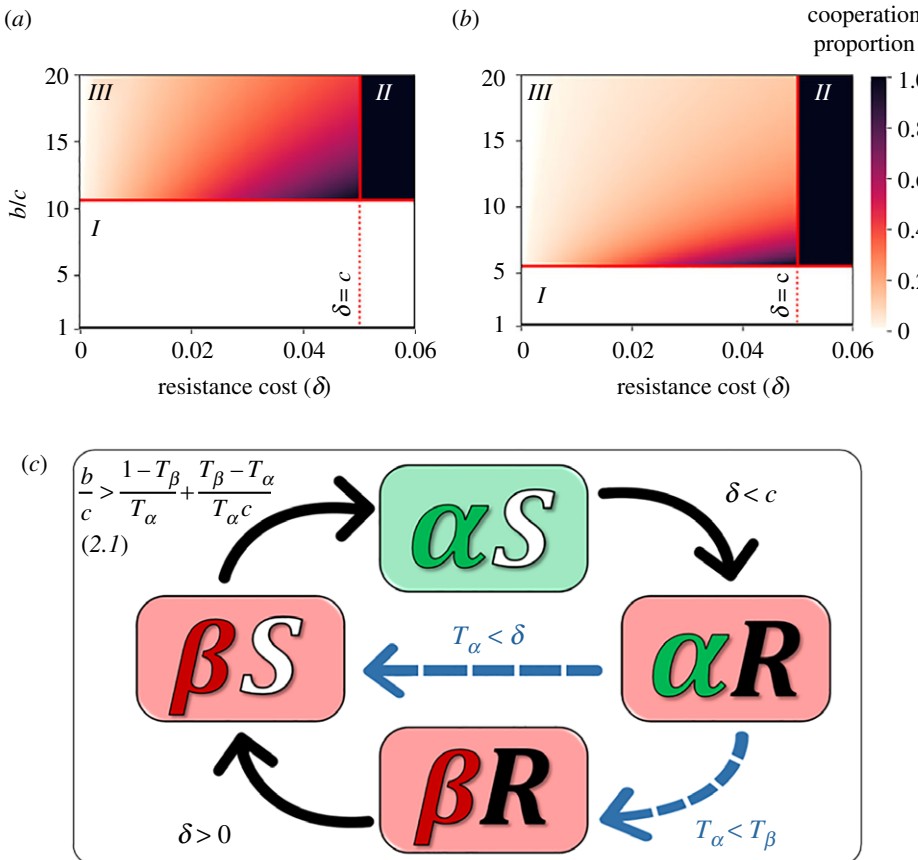

**Figure 2.** Cooperation can be maintained at intermediate levels even in the presence of host resistance to the microbial effect. (*a,b*) The expected proportion of cooperative hosts ($\alpha S$; colour coded), as a function of $b/c$ (*y*-axis) and $\delta$ (*x*-axis) for $c = 0.05$, $T_\beta = 0.25$, and (*a*) $T_\alpha = 0.75 \cdot T_\beta$, (*b*) $T_\alpha = 0.9 \cdot T_\beta$. Cooperation goes extinct when below the horizontal line representing condition (2.1) (area *I*, white). Above that threshold, cooperation can either go to fixation (when $\delta > c$, area *II*, black), or be maintained at intermediate levels (when $\delta < c$, area *III*). The expected proportions of all four host types are shown in electronic supplementary material, figure S3. (*c*) Rock–paper–scissors game of cooperation. Illustrated are the conditions that allow the invasion of a rare type to a population dominated by another type, based on invasion analysis (electronic supplementary material, S2). For example, if $\delta < c$, then $\alpha R$ hosts can invade $\alpha S$ populations, and if $T_\alpha < T_\beta$ then $\alpha R$ populations can be invaded by $\beta R$ hosts. Altogether, cooperation will be maintained as long as all the conditions represented by black arrows are satisfied, in addition to at least one of the conditions represented by dashed blue arrows.

and cooperation can evolve under a wide parameter range and be maintained at intermediate levels (area *III* in figure 2*a,b*). The proportion of cooperators at this polymorphic equilibrium increases with the cost of resistance ($\delta$), but is bounded: cooperators can reach up to $\delta/c$ from the proportion of hosts carrying allele $S$ in the population. Counterintuitively, above the $b/c$ threshold (of condition (2.1)) the proportion of cooperators at equilibrium decreases with $b/c$, due to the evolution of resistance.

Cooperation in our model can evolve even under transmission disadvantage. In that case, cooperation cannot evolve by infectivity alone. Kin selection among hosts is not a major factor either, as we here consider fully mixed populations. It is rather kin selection at the microbial level that enables the evolution of cooperation: microbe-induced cooperation evolves due to the ability to preferentially direct the cooperation benefit towards other (future) cooperators, *even in fully mixed populations*.

To identify the conditions that allow polymorphism, we investigated the stability of the four trivial equilibria (namely, the fixations of the four host types). Invasion analysis (detailed in electronic supplementary material, S2) revealed that when $0 < \delta < c$, cooperation evolves whenever:

$$T_\alpha < \max (T_\beta, \delta). \tag{2.3}$$

When (2.1) and (2.3) are satisfied and $0 < \delta < c$, no equilibrium that involves extinction of some of the host types, is stable. Under these conditions, cooperation is maintained at rock–paper–scissors dynamics (figures 2*c* and 3). Condition (2.3) is somewhat counterintuitive: for example, when $\alpha$ microbes have a transmission disadvantage ($T_\alpha < T_\beta$), it can facilitate the evolution of cooperation by hindering the evolution of resistance.

Interestingly, we found that the behaviour of the polymorphic system is oscillatory, and the population can either converge with oscillations towards the equilibrium (figure 3*a,b*) or oscillate chaotically around the equilibrium (figure 3*c,d*), depending on the different parameters and on the initial conditions of the population (electronic supplementary material, figure S4). In a population undergoing such chaotic oscillations, cases of near-fixation of one of the types are frequent. The behaviour of a finite population undergoing similar dynamics is thus intriguing: could cooperation still be maintained?

## (c) Stochastic models

So far, we have considered an infinite population where the dynamics are deterministic. Here, we study the evolution of microbe-induced cooperation in finite populations, subject to stochastic effects, and consider mutations, imperfect vertical transmission, and multiple interactions per generation.

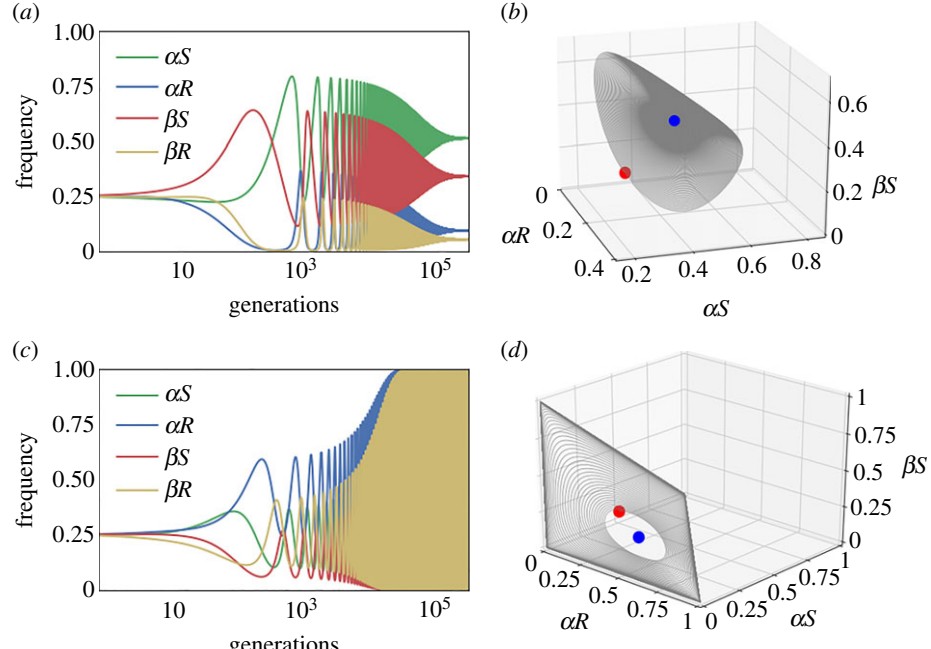

**Figure 3.** Oscillations of cooperation can either converge or diverge. We plot the frequencies of the four host types in the population with time (a, c) and in a three-dimensional plane (b,d) for $c = 0.05$, $\delta = 0.03$, $T_\beta = 0.25$, $T_\alpha = 0.9 T_\beta$ (based on iterations of equations (S1–S4) of electronic supplementary material, S1). (a,b) $b/c = 6$. The population spirals and converges to the polymorphic equilibrium. (c,d) $b/c = 10$. The population oscillates and diverges. The red dots in panels (b,d) represent the initial state of the population, with equal proportions of the four host types, and the blue dots represent the system's polymorphic equilibrium.

The stochastic simulations model populations of 10 000 interacting hosts (see simulation workflow in electronic supplementary material, S4). Similar to the analytic model, each host carries microbes of type $\alpha$ or $\beta$ and allele $R$ or $S$, and horizontal transmission of the microbes can occur during interactions. Each generation, hosts are paired randomly $K$ times, and in each pairing one interaction takes place between the two members. We start with a fully mixed and mutation-free population with one interaction per generation and perfect vertical transmission, corresponding to the deterministic model.

The simulation results strongly agree with the deterministic fully mixed model with regard to the conditions that lead to the fixation or extinction of cooperation, but there is a difference with regard to the conditions that allow polymorphism. While in the deterministic model we found long-term polymorphism, in the finite population cooperation goes extinct in many of the parameter sets (compare area **III** of figure 2a,b to area **III** of figure 4a).

Two mechanisms that can maintain polymorphism in an oscillating population are mutations and spatial structure [71–74]: mutations keep generating hosts and microbes of all types in the population, and by that rescue rare and extinct types; spatial structure limits both dispersal and interaction to the local scale, thus decreasing the strength of competition between the different host types—even when a type is common in the population, it is not common in every patch. As a result, the strength of oscillations decreases and all host types are maintained for a long time.

We therefore extended the simulation to account for both mutations and spatial structure (see details in electronic supplementary material, S4). Mutations were modelled as a random change (with rate $\mu$) in an offspring allele or/and microbe type, relative to its parent. Spatial structure was modelled similarly to [75], using a two-dimensional-lattice of size $100 \times 100$, where each site is inhabited by one host.

Differently from the fully mixed model, in the spatially structured population the interactions occur only between hosts inhabiting neighbouring sites, and selection is local as well. After the hosts interact, reproduction takes place and each site is inhabited by an offspring of a parent from the neighbourhood ($3 \times 3$ sites, or less if adjacent to the border), chosen with probability proportional to the parent's fitness. The offspring carries the same allele and microbe type as its parent, to the point of mutations.

We find that both mutations and spatial structure dramatically widen the range of parameters that allow the maintenance of cooperation in polymorphism, when $\delta < c$ and (2.1) is satisfied. Most of the effect is because both mechanisms reduce the probability of stochastic extinction of genotypes (compare area **III** of figure 4b and c to a; see also electronic supplementary material, video file, displaying the dynamics of one spatially structured population). However, even in populations with mutations, spatial structure yields slightly higher proportions of cooperation, probably due to the additional effect of kin selection among hosts (compare area III of figure 4d,c; see further analysis of the effects of host and microbe kin selection in electronic supplementary material, figure S5). All the stochastic simulations (with or without mutation and spatial structure) were also consistent with the deterministic results regarding the parameter range where cooperation goes extinct or fixates.

We find that without mutations, the proportion of cooperators is not monotonic in $\delta$ (see area *III* in figure 4a,b). When $\delta$ increases towards $c$, $\alpha R$ takes over the population frequently, while when $\delta$ decreases towards 0, fixation of $\beta S$ becomes common (see electronic supplementary material, S2 and figure S9). When allowing mutations, this pattern vanishes (compare figure 4a to c, and b to d).

We further examined the effect of imperfect vertical transmission, modelled similar to [29] (and its electronic supplementary material): with probability $\rho$ the offspring

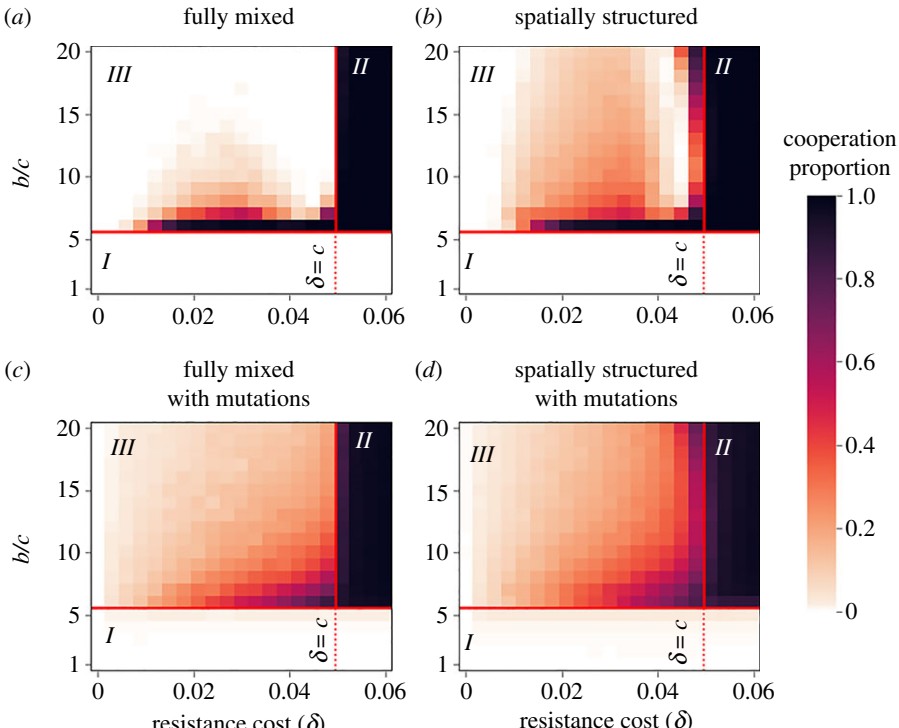

**Figure 4.** Mutations and spatial structure help maintain cooperation in the face of host resistance in finite populations. The proportion of cooperators after up to 5000 generations is plotted as a function of the $b/c$ ratio on the $y$-axis and $\delta$ on the $x$-axis. The colour of each square represents the average of 200 stochastic simulation runs. Panels ($a,c$) show the results of fully mixed populations, while panels ($b,d$) show the results of spatially structured populations. Panels ($a,b$) show the results without mutations, while panels ($c,d$) show the results with mutation rates of $\mu = 10^{-4}$ in all directions ($\alpha \leftrightarrow \beta$ and $S \leftrightarrow R$). Consistent with the analytic results, we find that cooperation goes extinct if (2.1) is not maintained, and cooperation fixates if (2.1) is satisfied and $\delta > c$. Both mutations and spatial structure significantly widen the parameter range allowing the maintenance of cooperation in polymorphism. Simulation parameters: $T_\beta = 0.25$, $T_\alpha = 0.9 T_\beta$, $c = 0.05$. All simulations used one interaction per generation, perfect vertical transmission of both alleles and microbes, and were initialized with equal proportions of the four host types. The results were robust to different mutation rates of $\mu = 10^{-3}$ and $\mu = 10^{-5}$ (electronic supplementary material, figure S6). The presented intermediate averages in area *III* usually represent population polymorphisms, apart from panel ($a$) (electronic supplementary material, figure S7). Stopping criteria is detailed in electronic supplementary material, S4.1. The results were robust to an alternative stopping threshold of 10 000 generations (electronic supplementary material, figure S8).

inherits its parent's microbes, and with probability $1-\rho$ it inherits microbes from a random host from the parent generation (see further details in electronic supplementary material, S4). We found that cooperation can evolve and be maintained even when the vertical transmission is far from perfect (electronic supplementary material, figure S10). Finally, we extended the model to account for multiple interactions ($K$) per host per generation and found qualitatively similar results (electronic supplementary material, figures S11 and S12).

## 4. Discussion

Our results demonstrate that cooperation induced by microbes can evolve in a host population under a wide range of parameters including the case where the hosts coevolve and acquire resistance to the microbial induction of cooperation. Although cooperative hosts bear an inherent disadvantage, the host–microbe coevolution generates a rock–paper–scissors dynamics in the population that enables the evolution and maintenance of cooperation. In addition, we find that the cost of resistance to the microbial effect ($\delta$) is a crucial factor. If this cost is higher than the cost of cooperation ($c$) then cooperation can fixate in the host population, and if not, cooperation can be maintained at an intermediate proportion through oscillatory rock–paper–scissors dynamics. We find that in finite-size populations, cooperation can also be

maintained in polymorphism, in the presence of either spatial structure or mutations of the microbes and the hosts.

Our framework can be extended in several directions. We currently model a binary microbe (either $\alpha$ or $\beta$) and host alleles (either $S$ or $R$). In natural populations, host behaviour might be affected by several loci in the genome, and by the composition of the microbial community, including the cases of co-infection with multiple microbes with different effects. In addition, the host behaviour can be modelled as a continuous trait, where the level of cooperation is varied, or condition-dependent. Other interesting host–microbe dynamics can arise when the cooperative behaviour is applied only under certain circumstances, e.g. when the hosts are under stress, similarly to other stress-induced behaviours [76–79]. It would also be interesting to examine the evolution of the rates of microbe horizontal and vertical transmission, in the context of microbe-induced cooperation. Once cooperation is common in the population, it can further affect the future evolution of the population [80,81].

There is already some evidence suggesting that microbes can affect cooperative behaviour in their host. Among eukaryotic hosts, *Lactobacillus* is a promising candidate: accumulating evidence demonstrates that *Lactobacillus* decreases stress-induced anxiety-like behaviour, potentially increasing the tendency of its host to interact with other individuals, and promoting cooperative behaviour [82–85]. In plants, carbon sharing among trees is, at least partially,

mediated by mycorrhizal fungi that form networks connecting neighbouring tree roots [86,87]. This behaviour can be considered as cooperation among trees, induced by their fungi. And if considering bacteria as hosts inhabited by mobile genetic elements, it was shown that genes responsible for public goods secretion are prevalent on mobile elements that can be transmitted both horizontally and vertically between host cells [68,69]. Our results suggest that such effects may be common in many other systems as well, and call for experimental tests.

This work can also be viewed in a different context of gene-culture coevolution [88,89]. Similar to microbes, culture also affects behaviour and specifically the tendency to cooperate, and can be transmitted both vertically and horizontally. Parents teach their offspring cultural–behavioural traits, but these traits can also be transmitted horizontally, through imitation of interaction partners [90]. Moreover, like microbes, culture interacts and coevolves with the genome, and genes can also affect the tendency of individuals to follow cultural rules or not (resistance) [91,92]. Differently from our model, a resistant genotype would not express cooperative behaviour and thus would probably not transfer it through imitation.

This study suggests that microbes may have a significant role in shaping their hosts' behaviour. It also demonstrates how a conflict between hosts and their microbes, portrayed by the ability of the hosts to evolve resistance to the microbial effect, can lead the population to a rock–paper–scissors game. This game enables long-term maintenance of cooperation, at intermediate levels. These results strengthen the theory of microbe-induced cooperation, and may help explain occurrences of cooperation that are difficult to explain by current theories. Our results provide verifiable predictions that can be tested in future experimental efforts: first, that altering the composition of microbial communities (e.g. by antibiotics [93–95], probiotics [96,97], pesticides [98], and herbicides [99]) may affect the host's social behaviour; and second, that polymorphism with respect to cooperative behaviour can be expected in natural populations, originating in a conflict between host genes and microbes.

Data accessibility. The simulation code is available from online service Zenodo, with doi:10.5281/zenodo.3629891.

Authors' contributions. O.L.-E. and L.H. designed the study and formulated the model. O.L.-E. derived the analytical equations and wrote the simulation code. O.L.-E. and L.H. analysed the results and wrote the manuscript.

Competing interests. We declare we have no competing interests.

Funding. This project was supported in part by the Clore Foundation Scholars Programme (OLE) and by the Israeli Science Foundation 2064/18 (LH).

Acknowledgements. We thank David Burstein for sharing computational resources and Ranit Aharonov for comments on the manuscript.

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
