## [Reviewer comments · Proceedings of the Royal Society B: Biological Sciences]

Review History

RSPB-2019-1558.R0 (Original submission)

Review form: Reviewer 1

Recommendation

Accept with minor revision (please list in comments)

Scientific importance: Is the manuscript an original and important contribution to its field?

Excellent

General interest: Is the paper of sufficient general interest?

Excellent

Quality of the paper: Is the overall quality of the paper suitable?

Excellent

Is the length of the paper justified?

Yes

Should the paper be seen by a specialist statistical reviewer?

No

Do you have any concerns about statistical analyses in this paper? If so, please specify them explicitly in your report.

No

It is a condition of publication that authors make their supporting data, code and materials available - either as supplementary material or hosted in an external repository. Please rate, if applicable, the supporting data on the following criteria.

Is it accessible?

Yes

Is it clear?

N/A

Is it adequate?

N/A

Do you have any ethical concerns with this paper?

No

Comments to the Author

The authors investigate the evolution of a microbe that can induce cooperation in its hosts and the evolution of host resistance/susceptibility to microbial influences. They find cases where cooperation is present in some stable, positive fraction of the population as well as cases where the levels of cooperation show cyclical dynamics with other combinations of host and microbial genotypes. This builds on previous work investigating the evolution of microbe-induced cooperation where hosts were unable to resist their microbes effects. Adding hosts resistance adds an important layer of biological realism and produces interesting new dynamics. The question of when hosts would allow themselves to be "coerced" into cooperating is also an extremely interesting one. The particular model the authors use is a good way to investigate the question. There are some minor things that could be changed, mainly to improve clarity.

I think the most important issue for clarity has to do with the order of events when hosts interact. In the model, horizontal transmission occurs when hosts meet to play a prisoner's dilemma game. It is not clear from the main text whether transmission occurs before or after play, although equations 1 - 4 in the supplement suggest that transmission occurs after play (or, if it occurs before play, that any newly-acquired microbe does not affect the host's behavior). I think this needs to be made explicit in the main text to avoid confusion with another possible model of how relatedness between microbes could allow the spread of cooperation - if microbes were to transmit and influence their new host's behavior prior to play, cooperators could potentially increase their chances of interacting with another cooperator.

Results:

Figure 1 is very clear and helpful for understanding the model.

Figure 2: Would it be possible to have a supplementary figure showing the expected fractions of other host-microbe combinations? It would be interested to see how these change over parameter space as well.

Figure 3: These plots are extremely helpful for understanding the dynamics of the system. Having the start point highlighted is really useful for visualizing the direction of evolution. Would it be possible to highlight the location of the equilibrium as well?

Figure 4: I am intrigued by the shape of the region where cooperation does not go extinct in a and b. Is there any intuition about why it looks like this?

Discussion:

Lines 330 – 338: I am excited about the possibility of this model being applicable to gene-culture coevolution. The way cultural transmission is described here, I am not sure if it exactly fits the model. If individuals must see their interaction partner exhibit a behavior to acquire it horizontally, then “ α R” individuals seem like they will transmit the “ β ” cultural trait, rather than “ α .”

Methods:

Supplementary equations 1 – 4 provide a lot of insight into the model, and it would be great if they or their simplified versions were in the methods section.

What fractions of host types were the simulations initialized with? I imagine the oscillations in the fraction of infected hosts are not so regular that this is a problem, but does the proportion of cooperators at the end of 100 simulations cover the range of variation in the fraction of cooperators?

Lines 374-377: I don’t understand how interactions and transmission work in the simulations with spatial structure. What is the impact of choosing an individual to initiate an interaction? What is the fitness of an individual who, by chance, never interacts with anyone? How do transmission and microbe effects work when there are potentially multiple interactions within a generation? For example, can an α S host interact with a β S neighbor, become infected, and then, in its interaction with a second neighbor, re-acquire an α microbe? Does this host behave non-cooperatively in its interaction with the second neighbor?

Supplement:

I think the authors' decision to show only the extended model is very reasonable. For readers looking for the simplified model, it would be good to state that only the extended model will be discussed in S1 and S2 and explain how readers can get the results for the simplified model from the extended.

The explanations of how the Jacobians were found are excellent.

It would be great if the descriptions of the eigenvalues for invasion gave some explanation about the role of these values in the analysis and what each corresponds to.

Lines 78-80: This sentence is confusing.

Some explanation of how the intersection of conditions was found that allows the coexistence of multiple types would be wonderful.

S3.2 explains the polymorphic equilibria and the dynamics of the system really well.

I am curious about the conditions where the fractions of hosts of each type oscillate vs. reach equilibrium. Would it be possible to discuss which initial values lead to each outcome? Or perhaps to show a sample plot highlighting the initial fractions of each host type that lead to oscillation vs. equilibrium?

The supplementary video was extremely helpful for visualizing the dynamics of the system. I couldn’t find a reference to it in the main text, and I think one would be great.

Review form: Reviewer 2

Recommendation

Major revision is needed (please make suggestions in comments)

Scientific importance: Is the manuscript an original and important contribution to its field?

Acceptable

General interest: Is the paper of sufficient general interest?

Acceptable

Quality of the paper: Is the overall quality of the paper suitable?

Acceptable

Is the length of the paper justified?

Yes

Should the paper be seen by a specialist statistical reviewer?

No

Do you have any concerns about statistical analyses in this paper? If so, please specify them explicitly in your report.

No

It is a condition of publication that authors make their supporting data, code and materials available - either as supplementary material or hosted in an external repository. Please rate, if applicable, the supporting data on the following criteria.

Is it accessible?

N/A

Is it clear?

N/A

Is it adequate?

N/A

Do you have any ethical concerns with this paper?

No

Comments to the Author

This paper presents a model of cooperation (altruism in the inclusive fitness sense, modelled as a 2-player single-shot Prisoner's Dilemma) is induced in hosts through transmitted microbes. This builds on a model presented in a prior paper published by the authors: Lewin-Epstein O., Aharonov R., Hadany L. 2017 Microbes can help explain the evolution of host altruism. Nature communications 8, 14040 . That paper shows conditions under which a cooperative trait induced by horizontally transmitted microbes can spread, even when the hosts themselves are not related (in effect both because of horizontal transmission through infection and the establishment of relatedness at the locus for cooperation carried by the microbes).

The advance in the present paper is to now consider that hosts may be able to carry a gene that confers resistance to the effects of the microbes. The results are that this leads to rock-paper-scissors dynamics in terms of invasibility. Even when the cost of resistance is less than the cost of the induced cooperation in the Prisoner's Dilemma, a polymorphic equilibrium level of cooperation can result.

The results are potentially interesting theoretically and empirically. However, their presentation and interpretation needs to be strengthened:

1. Theoretically, the selective forces favouring cooperation in the model are not well explained. What exactly is the role of relatedness, both between microbes, and between hosts? What is the role of infectivity (the transmission rates)? Can you disentangle these?
2. The paper needs to connect more with the literature on horizontal transmission of cooperative traits through plasmids. In particular, the paper "Mc Ginty SE, Lehmann L, Brown SP, Rankin DJ. 2013 The interplay between relatedness and horizontal gene transfer drives the evolution of plasmid-carried public goods. Proc R Soc B 280: 20130400" would help to explain the effects of relatedness and infectivity. There are also many other papers in this area, and these should be discussed in the introduction of the paper.
3. The model relies on a dyadic interaction, where crucially horizontal transfer occurs immediately after the Prisoner's Dilemma game, between the same pair of individuals. I suspect that this strongly favours the spread of cooperation compared to the case where transmission is not directly coupled to the Prisoner's Dilemma game. How realistic empirically is the scenario that microbial exchange occurs during the Prisoner's Dilemma interaction?
4. What are the effects in the model of relaxing the assumption in 3. ?
5. Many interactions empirically involve n individuals rather than dyads. Would the mechanism of microbial transmission still be expected to work in that case?
6. Are there any concrete empirical examples of microbes inducing cooperation in their hosts?
7. The paper ends by stating that it makes precise testable predictions. The discussion should be more explicit about what exactly these are.

Review form: Reviewer 3

Recommendation

Accept with minor revision (please list in comments)

Scientific importance: Is the manuscript an original and important contribution to its field?

Good

General interest: Is the paper of sufficient general interest?

Good

Quality of the paper: Is the overall quality of the paper suitable?

Excellent

Is the length of the paper justified?

Yes

Should the paper be seen by a specialist statistical reviewer?

Yes

Do you have any concerns about statistical analyses in this paper? If so, please specify them explicitly in your report.

No

It is a condition of publication that authors make their supporting data, code and materials available - either as supplementary material or hosted in an external repository. Please rate, if applicable, the supporting data on the following criteria.

Is it accessible?

No

Is it clear?

No

Is it adequate?

No

Do you have any ethical concerns with this paper?

No

Comments to the Author

The manuscript investigates the interesting idea that microbes could manipulate the cooperation of their hosts with other hosts. The authors take a coevolutionary approach to understand the stability of microbe-mediated host cooperation by including hosts that are susceptible and resistant to manipulation and microbes that do and do not manipulate hosts. The authors find that it is possible for microbe-mediated cooperation to persist in populations across a wide range of conditions. They also find that the inclusion of mutation and spatial structure increase the parameter space in which cooperation persists.

These findings are useful because they could explain the presence of cooperation in cases where it may not otherwise have been expected based on current theory. The manuscript is novel in this way.

The manuscript will be of interest to those researching cooperation, social behaviour in general, evolution and microbe-host interactions. It is also well-written and the logic is clear. I also appreciate the discussion towards the end, acknowledging what the models do not incorporate, e.g. the composition of the microbial community in the host. It will be interesting to consider how co-infection of a manipulating microbe with its non-manipulating counterpart will affect dynamics. Presumably, if this non-manipulating microbe can benefit from the cooperation induced by the manipulating microbe, it will act as a 'cheat', which could affect the conditions under which cooperation will persist.

Specific points:

Line 39: As far as I understand currently-accepted terminology, 'cooperation' can include interactions where there is no cost, while your definition and model explicitly state a >0 cost. This is defined as altruism. You might want to mention this.

Line 278: "maintaining" should be "maintain".

Line 383, "where" should be "were".

Decision letter (RSPB-2019-1558.R0)

13-Aug-2019

Dear Dr Hadany:

I am writing to inform you that your manuscript RSPB-2019-1558 entitled "Host-microbiome coevolution promotes cooperation in a rock-paper-scissor dynamic" has, in its current form, been rejected for publication in Proceedings B.

This action has been taken on the advice of referees and the Associate Editor, who have recommended that substantial revisions are necessary. With this in mind we would be happy to consider a resubmission, provided the comments of the referees and the Associate Editor are fully addressed. However please note that this is not a provisional acceptance.

Sincerely,
 Professor Hans Heesterbeek
 mailto: proceedingsb@royalsociety.org

Associate Editor
 Board Member: 1
 Comments to Author:

This ms presents a host-microbe model whereby one of two competing strains of the microbe induce an altruistic act in its host, provided that the host is of one of two genotypes; the other host genotype is resistant towards this manipulation by the microbe. The altruistic interaction is the only opportunity for the microbe to be transmitted horizontally; they also transmit vertically with 100 % fidelity. For a broad range of parameters, the altruism-inducing microbe strain can invade, and thus altruistic behavior can persist. Notably, a protected polymorphism in both species, with dampening or persistent oscillations can be maintained even if the altruism-inducing microbe strain has a disadvantage in horizontal transmission; this is because the successfully transmitted propagule enjoy the extra advantage in vertical transmission because the latter is proportional to host fitness and the newly infected host simultaneously receives the fitness benefit of altruism.

All three reviewers found the topic of the paper interesting, but while reviewers 1 and 3 were positive, reviewer 2 was more critical and raised important issues concerning the novelty and biological generality and relevance of this work.

Concerning the novelty, at an abstract level the model seems similar to models of plasmid-encoded cooperation between microbes. Indeed, the authors bring up such a case as one of the examples relevant for their model (l. 320-2). As reviewer points out, there is literature on the conditions for the evolution of cooperation under this kind of scenario. The authors should make

a clear case of how this paper is novel compared to that literature both in the biological scenario where it might apply and in the results/predictions.

Concerning the biological relevance, reviewer 2 questions the assumption that horizontal transmission is completely tied to the altruistic act. Thus, it is not possible to say to what degree the advantage is due to inducing transmission-enabling host contact versus inducing the actual altruism. This seems rather biologically unrealistic, and works strongly in favor of altruism in this model. I agree that the authors should explore the consequences of relaxing this assumption. One way would be to analyze a version where hosts infected by microbe "beta" also initiate interactions that create the same opportunity for the microbe transmission, but without affecting host fitness ($b = c = 0$). The reviewer would also like to know how the model would generalize to each individual participating in multiple interactions (rather than just one per generation). Possibly, an interaction with n individuals with given values of b , c , and T_{α} , T_{β} would yield similar results as one interaction with those parameters multiplied by n , but the authors should explore this. From my side, I find the assumptions about microbe transmission rather peculiar. The hosts cannot clear the microbe and vertical transmission is perfect, so there are no uninfected hosts. Yet, once microbe alpha is transmitted to the host carrying microbe beta or vice versa, it results in elimination of the other strain, so no persistent multiple infection is possible. So, in effect, the host cannot get rid of neither microbe alpha nor beta, but microbe alpha infecting host already infected with beta eliminates the latter and vice versa (with some probability < 1). At least a preliminary investigation of relaxing these assumptions – allowing for vertical transmission with probability < 1 and/or for simultaneous infection with the two microbes should be carried out. The latter case might in particular be interesting, because it would generate situations where microbe alpha induces an altruistic act, but it happens to be microbe beta that is transmitted and thus profits from the altruism. Without considering such broader – and possibly biologically more realistic – assumptions, the statements that the model explains cooperation "in a wide variety of organisms" (l. 34-35 or 303-304) is simply not justified. It may be true that this holds under a broad range of parameters, but in the framework of a highly specific model. Reviewers 1 and 2 also see potential to improve the presentation. Notably, the model results could be better intuitively explained and dissected. In addition to the specific example mentioned in the reviews, it is not clear to me to what degree the more favorable conditions for altruism in spatially viscous population in the simulations are due to kin selection in the host versus the microbe. One way of resolving it would be to run some simulations in which either the host or the microbe genotypes are spatially randomized every generation. The reviewers point to several other potentially improvements and issues that the authors should carefully consider.

Additional comments:

The ability to manipulate host behavior is clearly not limited to microbes. Nonmicrobial parasites, commensals or symbionts can also be transmitted horizontally, what is said in lines 76-83 clearly applies to many multicellular parasites. OK, I see the authors say in passing in the Discussion that their results apply to those other taxa, but why not to state so from the beginning rather than insisting first that microbes are somehow different?

l. 320-322: This does not seem to be an example of microbes manipulating their multicellular hosts, but of a horizontally transferable genetic element affecting the cooperative behavior of their microbial carriers. So it does not fit the preceding sequence. On the other hand it seems to me that it does fit the general logic of the model. It may actually be the most biologically plausible case where the basic premise of this model applies (although the specific assumptions about transmission might still not be plausible). So this is another reason to specify the idea in more general terms of horizontally and vertically transmitted infectious agents.

l. 180-188: an important piece of information missing in that otherwise clear intuitive explanation is that the polymorphism in the host is maintained by a balance between its disadvantage when infected with alpha and its advantage when infected with beta. This directly explains the

equilibrium property in l. 179 – at this frequency of alpha infection the fitness of R and S hosts is equal.

l. 323: the authors should follow the rules of biological nomenclature like capitalizing genus names

Reviewer(s)' Comments to Author:

Referee: 1

Comments to the Author(s)

The authors investigate the evolution of a microbe that can induce cooperation in its hosts and the evolution of host resistance/susceptibility to microbial influences. They find cases where cooperation is present in some stable, positive fraction of the population as well as cases where the levels of cooperation show cyclical dynamics with other combinations of host and microbial genotypes. This builds on previous work investigating the evolution of microbe-induced cooperation where hosts were unable to resist their microbes effects. Adding hosts resistance adds an important layer of biological realism and produces interesting new dynamics. The question of when hosts would allow themselves to be “coerced” into cooperating is also an extremely interesting one. The particular model the authors use is a good way to investigate the question. There are some minor things that could be changed, mainly to improve clarity.

I think the most important issue for clarity has to do with the order of events when hosts interact. In the model, horizontal transmission occurs when hosts meet to play a prisoner's dilemma game. It is not clear from the main text whether transmission occurs before or after play, although equations 1 – 4 in the supplement suggest that transmission occurs after play (or, if it occurs before play, that any newly-acquired microbe does not affect the host's behavior). I think this needs to be made explicit in the main text to avoid confusion with another possible model of how relatedness between microbes could allow the spread of cooperation – if microbes were to transmit and influence their new host's behavior prior to play, cooperators could potentially increase their chances of interacting with another cooperator.

Results:

Figure 1 is very clear and helpful for understanding the model.

Figure 2: Would it be possible to have a supplementary figure showing the expected fractions of other host-microbe combinations? It would be interested to see how these change over parameter space as well.

Figure 3: These plots are extremely helpful for understanding the dynamics of the system. Having the start point highlighted is really useful for visualizing the direction of evolution. Would it be possible to highlight the location of the equilibrium as well?

Figure 4: I am intrigued by the shape of the region where cooperation does not go extinct in a and b. Is there any intuition about why it looks like this?

Discussion:

Lines 330 – 338: I am excited about the possibility of this model being applicable to gene-culture coevolution. The way cultural transmission is described here, I am not sure if it exactly fits the model. If individuals must see their interaction partner exhibit a behavior to acquire it horizontally, then “ α R” individuals seem like they will transmit the “ β ” cultural trait, rather than “ α .”

Methods:

Supplementary equations 1 – 4 provide a lot of insight into the model, and it would be great if they or their simplified versions were in the methods section.

What fractions of host types were the simulations initialized with? I imagine the oscillations in the fraction of infected hosts are not so regular that this is a problem, but does the proportion of cooperators at the end of 100 simulations cover the range of variation in the fraction of cooperators?

Lines 374-377: I don't understand how interactions and transmission work in the simulations with spatial structure. What is the impact of choosing an individual to initiate an interaction? What is the fitness of an individual who, by chance, never interacts with anyone? How do transmission and microbe effects work when there are potentially multiple interactions within a generation? For example, can an α S host interact with a β S neighbor, become infected, and then, in its interaction with a second neighbor, re-acquire an α microbe? Does this host behave non-cooperatively in its interaction with the second neighbor?

Supplement:

I think the authors' decision to show only the extended model is very reasonable. For readers looking for the simplified model, it would be good to state that only the extended model will be discussed in S1 and S2 and explain how readers can get the results for the simplified model from the extended.

The explanations of how the Jacobians were found are excellent.

It would be great if the descriptions of the eigenvalues for invasion gave some explanation about the role of these values in the analysis and what each corresponds to.

Lines 78-80: This sentence is confusing.

Some explanation of how the intersection of conditions was found that allows the coexistence of multiple types would be wonderful.

S3.2 explains the polymorphic equilibria and the dynamics of the system really well.

I am curious about the conditions where the fractions of hosts of each type oscillate vs. reach equilibrium. Would it be possible to discuss which initial values lead to each outcome? Or perhaps to show a sample plot highlighting the initial fractions of each host type that lead to oscillation vs. equilibrium?

The supplementary video was extremely helpful for visualizing the dynamics of the system. I couldn't find a reference to it in the main text, and I think one would be great.

Referee: 2

Comments to the Author(s)

This paper presents a model of cooperation (altruism in the inclusive fitness sense, modelled as a 2-player single-shot Prisoner's Dilemma) is induced in hosts through transmitted microbes. This builds on a model presented in a prior paper published by the authors: Lewin-Epstein O., Aharonov R., Hadany L. 2017 Microbes can help explain the evolution of host altruism. *Nature communications* 8, 14040. That paper shows conditions under which a cooperative trait induced by horizontally transmitted microbes can spread, even when the hosts themselves are not related (in effect both because of horizontally transmission through infection and the establishment of relatedness at the locus for cooperation carried by the microbes).

The advance in the present paper is to now consider that hosts may be able to carry a gene that confers resistance to the effects of the microbes. The results are that this leads to rock-paper-scissors dynamics in terms of invasability. Even when the cost of resistance is less than the cost of the induced cooperation in the Prisoner's Dilemma, a polymorphic equilibrium level of cooperation can result.

The results are potentially interesting theoretically and empirically. However, their presentation and interpretation needs to be strengthened:

1. Theoretically, the selective forces favouring cooperation in the model are not well explained. What exactly is the role of relatedness, both between microbes, and between hosts? What is the role of infectivity (the transmission rates)? Can you disentangle these?
2. The paper needs to connect more with the literature on horizontal transmission of cooperative traits through plasmids. In particular, the paper "Mc Ginty SE, Lehmann L, Brown SP, Rankin DJ. 2013 The interplay between relatedness and horizontal gene transfer drives the evolution of plasmid-carried public goods. *Proc R Soc B* 280: 20130400" would help to explain the effects of relatedness and infectivity. There are also many other papers in this area, and these should be discussed in the introduction of the paper.
3. The model relies on a dyadic interaction, where crucially horizontal transfer occurs immediately after the Prisoner's Dilemma game, between the same pair of individuals. I suspect that this strongly favours the spread of cooperation compared to the case where transmission is not directly coupled to the Prisoner's Dilemma game. How realistic empirically is the scenario that microbial exchange occurs during the Prisoner's Dilemma interaction?
4. What are the effects in the model of relaxing the assumption in 3. ?
5. Many interactions empirically involve n individuals rather than dyads. Would the mechanism of microbial transmission still be expected to work in that case?
6. Are there any concrete empirical examples of microbes inducing cooperation in their hosts?
7. The paper ends by stating that it makes precise testable predictions. The discussion should be more explicit about what exactly these are.

Referee: 3

Comments to the Author(s)

The manuscript investigates the interesting idea that microbes could manipulate the cooperation of their hosts with other hosts. The authors take a coevolutionary approach to understand the stability of microbe-mediated host cooperation by including hosts that are susceptible and resistant to manipulation and microbes that do and do not manipulate hosts. The authors find that it is possible for microbe-mediated cooperation to persist in populations across a wide range of conditions. They also find that the inclusion of mutation and spatial structure increase the parameter space in which cooperation persists.

These findings are useful because they could explain the presence of cooperation in cases where it may not otherwise have been expected based on current theory. The manuscript is novel in this way.

The manuscript will be of interest to those researching cooperation, social behaviour in general, evolution and microbe-host interactions. It is also well-written and the logic is clear. I also appreciate the discussion towards the end, acknowledging what the models do not incorporate, e.g. the composition of the microbial community in the host. It will be interesting to consider how co-infection of a manipulating microbe with its non-manipulating counterpart will affect dynamics. Presumably, if this non-manipulating microbe can benefit from the cooperation induced by the manipulating microbe, it will act as a 'cheat', which could affect the conditions under which cooperation will persist.

Specific points:

Line 39: As far as I understand currently-accepted terminology, 'cooperation' can include interactions where there is no cost, while your definition and model explicitly state a >0 cost. This is defined as altruism. You might want to mention this.

Line 278: "maintaining" should be "maintain".

Line 383, "where" should be "were".

Author's Response to Decision Letter for (RSPB-2019-1558.R0)

See Appendix A.

RSPB-2019-2754.R0

Review form: Reviewer 1

Recommendation

Accept with minor revision (please list in comments)

Scientific importance: Is the manuscript an original and important contribution to its field?

Excellent

General interest: Is the paper of sufficient general interest?

Excellent

Quality of the paper: Is the overall quality of the paper suitable?

Excellent

Is the length of the paper justified?

Yes

Should the paper be seen by a specialist statistical reviewer?

No

Do you have any concerns about statistical analyses in this paper? If so, please specify them explicitly in your report.

No

It is a condition of publication that authors make their supporting data, code and materials available - either as supplementary material or hosted in an external repository. Please rate, if applicable, the supporting data on the following criteria.

Is it accessible?

N/A

Is it clear?

N/A

Is it adequate?

N/A

Do you have any ethical concerns with this paper?

No

Comments to the Author

I really like the authors' changes, which I think have made things a lot clearer. In particular I think the new model explanation is really clear and gives a very complete description of the model.

I do have a small comment about the effect of multiple interactions on fitness in the simulations, mainly the ones of spatially structured populations. Because the costs and benefits of multiple interactions aren't normalized by the number of interactions, the expected fitness of an individual interacting with the same distribution of partners can change depending on the number of partners it has. The authors recognize this and deal with it by dividing the cost of cooperation by the expected number of interactions in Figure S11.

This is great for the well-mixed simulations (where number of interactions is fixed). It's probably not a problem for the spatially structured simulations either, given how robust the simulations are to other changes in the model. However, the relationship between expected fitness and number of interactions might have an impact on hosts in these simulations, since hosts have a variable number of interactions. I think it's worth mentioning this in the text or possibly doing a quick check of what happens when the expected payoff isn't a function of the number of interactions, if that's not too much trouble.

A few other very minor things:

Line 115: I think "to the microbial effect" or similar probably needs to remain to clarify that "resistance" doesn't refer to resistance to infection.

Line 430: "considering bacteria as hosts that inhabit mobile genetic elements" should be "considering bacteria as hosts inhabited by mobile genetic elements"

General: "their host behavior" should be "their host's behavior"

Review form: Reviewer 2**Recommendation**

Accept as is

Scientific importance: Is the manuscript an original and important contribution to its field?

Good

General interest: Is the paper of sufficient general interest?

Good

Quality of the paper: Is the overall quality of the paper suitable?

Good

Is the length of the paper justified?

Yes

Should the paper be seen by a specialist statistical reviewer?

No

Do you have any concerns about statistical analyses in this paper? If so, please specify them explicitly in your report.

No

It is a condition of publication that authors make their supporting data, code and materials available - either as supplementary material or hosted in an external repository. Please rate, if applicable, the supporting data on the following criteria.

Is it accessible?

N/A

Is it clear?

N/A

Is it adequate?

N/A

Do you have any ethical concerns with this paper?

No

Comments to the Author

The authors have modified the paper to address my concerns.

Decision letter (RSPB-2019-2754.R0)

24-Dec-2019

Dear Dr Hadany:

Your manuscript has now been peer reviewed and the reviews have been assessed by an Associate Editor. The reviewers' comments (not including confidential comments to the Editor) and the comments from the Associate Editor are included at the end of this email for your reference. As you will see, the reviewers are positive but one reviewer raised some issues with your manuscript and we would like to invite you to revise your manuscript to address them.

Research ethics:

Use of animals and field studies:

Please submit a copy of your revised paper within three weeks. If we do not hear from you

within this time your manuscript will be rejected. If you are unable to meet this deadline please let us know as soon as possible, as we may be able to grant a short extension.

Best wishes,
 Professor Hans Heesterbeek
 mailto: proceedingsb@royalsociety.org

Associate Editor Board Member

Comments to Author:

The reviewers are satisfied with the revisions done by the authors. However, reviewer 1 still raises one issue that the authors should address in discussion and possibly be making some exploratory simulations.

Reviewer(s)' Comments to Author:

Referee: 1

Comments to the Author(s).

I really like the authors' changes, which I think have made things a lot clearer. In particular I think the new model explanation is really clear and gives a very complete description of the model.

I do have a small comment about the effect of multiple interactions on fitness in the simulations, mainly the ones of spatially structured populations. Because the costs and benefits of multiple interactions aren't normalized by the number of interactions, the expected fitness of an individual interacting with the same distribution of partners can change depending on the number of partners it has. The authors recognize this and deal with it by dividing the cost of cooperation by the expected number of interactions in Figure S11.

This is great for the well-mixed simulations (where number of interactions is fixed). It's probably not a problem for the spatially structured simulations either, given how robust the simulations are to other changes in the model. However, the relationship between expected fitness and number of interactions might have an impact on hosts in these simulations, since hosts have a variable number of interactions. I think it's worth mentioning this in the text or possibly doing a quick check of what happens when the expected payoff isn't a function of the number of interactions, if that's not too much trouble.

A few other very minor things:

Line 115: I think "to the microbial effect" or similar probably needs to remain to clarify that "resistance" doesn't refer to resistance to infection.

Line 430: "considering bacteria as hosts that inhabit mobile genetic elements" should be "considering bacteria as hosts inhabited by mobile genetic elements"

General: "their host behavior" should be "their host's behavior"

Referee: 2

Comments to the Author(s).

The authors have modified the paper to address my concerns.

Author's Response to Decision Letter for (RSPB-2019-2754.R0)

See Appendix B.

Decision letter (RSPB-2019-2754.R1)

15-Jan-2020

Dear Dr Hadany

I am pleased to inform you that your manuscript entitled "Host-microbiome coevolution can promote cooperation in a rock-paper-scissor dynamic" has been accepted for publication in Proceedings B.

Open Access

Your article has been estimated as being 9 pages long. Our Production Office will be able to confirm the exact length at proof stage.

Paper charges

Sincerely,

Professor Hans Heesterbeek

Associate Editor:
Board Member
Comments to Author:
(There are no comments.)

Appendix A

Dear Editor,

We would like to thank the associate editor and the reviewers for their comprehensive and thorough review, which we believe guided us in significantly improving the manuscript.

Following the questions and suggestions, we made the following major changes:

1. We relaxed some of the assumptions and added new results showing that the model is robust to imperfect vertical transmission and multiple interactions per host per generation. We revised the manuscript to refer to this generalization of the initial model, and added new figures presenting the results (Figures S10 and S11).
2. We added a new analysis showing (Figure S5) the effect of breaking the association between the hosts and their microbes (by shuffling the microbes across the host population) and kin selection among microbes (by unlinking between the behavioral interaction and the horizontal transmission).
3. We added additional analysis of the results we obtained in the deterministic and stochastic models, to present a broader picture and improve clarity (Figures S3, S4, S7, S9).
4. We improved and clarified the presentation of the biological context and theoretical background.

We made numerous additional changes to the manuscript in order to address the editor's and the reviewers' comments and incorporate their suggestions. Below are the comments, each followed by our reply and the details of the revisions made to address the comment.

We found the comments very helpful and constructive. We believe we have addressed the editor's and the reviewers' comments and suggestions and we are confident that the new version of the manuscript is improved. We hope you find our manuscript suitable for publication in Proceedings of the Royal Society B.

Sincerely,

Lilach Hadany

Associate Editor

Board Member: 1

Comments to Author:

This ms presents a host-microbe model whereby one of two competing strains of the microbe induce an altruistic act in its host, provided that the host is of one of two genotypes; the other host genotype is resistant towards this manipulation by the microbe. The altruistic interaction is the only opportunity for the microbe to be transmitted horizontally; they also transmit vertically with 100 % fidelity. For a broad range of parameters, the altruism-inducing microbe strain can invade, and thus altruistic behavior can persist. Notably, a protected polymorphism in both species, with dampening or persistent oscillations can be maintained even if the altruism-inducing microbe strain has a disadvantage in horizontal transmission; this is because the successfully transmitted propagule enjoy the extra advantage in vertical transmission because the latter is proportional to host fitness and the newly infected host simultaneously receives the fitness benefit of altruism.

All three reviewers found the topic of the paper interesting, but while reviewers 1 and 3 were positive, reviewer 2 was more critical and raised important issues concerning the novelty and biological generality and relevance of this work.

Concerning the novelty, at an abstract level the model seems similar to models of plasmid-encoded cooperation between microbes. Indeed, the authors bring up such a case as one of the examples relevant for their model (l. 320-2). As reviewer points out, there is literature on the conditions for the evolution of cooperation under this kind of scenario. The authors should make a clear case of how this paper is novel compared to that literature both in the biological scenario where it might apply and in the results/predictions.

We thank the editor for this comment. We added a paragraph to the introduction, elaborating more on previous works related to mobile genetic elements and induction of public goods secretion, as well as explaining the novelty of our theory and model (L96):

“Previous studies examined the evolution of public goods genes that are encoded on mobile genetic elements that can be transmitted vertically and horizontally between bacteria ([67-70]). These studies explored the role of transmission and relatedness on the evolution of bacterial cooperation, both empirically and theoretically. Our work is different in several ways. First, we broaden the perspective and claim that in fact, the evolution of cooperation by horizontally-transmitted elements is relevant to

almost any species, via its microbiome, or any other symbiont. Second, we emphasize the link between social interaction and horizontal transmission, which we show, enables the evolution of cooperation under wide conditions. Last, we study host-microbe coevolution with respect to cooperation, by accounting for both microbial and host alleles.”

Concerning the biological relevance, reviewer 2 questions the assumption that horizontal transmission is completely tied to the altruistic act. Thus, it is not possible to say to what degree the advantage is due to inducing transmission-enabling host contact versus inducing the actual altruism. This seems rather biologically unrealistic, and works strongly in favor of altruism in this model. I agree that the authors should explore the consequences of relaxing this assumption. One way would be to analyze a version where hosts infected by microbe "beta" also initiate interactions that create the same opportunity for the microbe transmission, but without affecting host fitness ($b = c = 0$).

We agree, and the last assumption is already included in our model, as we now clarify in the revised manuscript. Hosts have the same number of interactions and the same opportunity for microbe transmission, irrespective of the microbe they carry. The only difference is the rates of horizontal transmission, and we focus on the case of transmission disadvantage to microbe α . Namely, we don't tie between cooperative interaction and horizontal transmission, but rather between any interaction and horizontal transmission. Microbe α only affects behavior during interaction, and microbe β doesn't. However note that a host carrying β can still benefit b from interacting with a cooperator.

We now emphasize this point in the Model section:

(L119): “Hosts interact in pairs, and microbes can affect their hosts behavior: microbes of type α increase the tendency of their hosts to cooperate during interaction, while microbes of type β don't have any effect over their host behavior.”

(L131):

“Note that the microbes do not affect the tendency of the hosts to interact. Thus, cooperators and non-cooperators take part in the same number of interactions.”

The reviewer would also like to know how the model would generalize to each individual participating in multiple interactions (rather than just one per generation). Possibly, an interaction with n individuals with given values of b , c , and T_{α} , T_{β} would yield similar results as one interaction with those parameters multiplied by n , but the authors should explore this.

We thank the reviewer and editor for pointing out this issue, which indeed was missing in the original draft. We extended our simulations to account for multiple interactions per generation, and were able to show that our model is robust to multiple interactions per generation – yielding qualitatively similar results. Indeed, as the editor mentions, by “correcting” the transmission rates and the cost of cooperation according to the number of interactions per generation, the obtained results are very similar.

We now refer to this point in the main text (L352):

“Finally, we extended the model to account for multiple interactions (K) per host per generation and found qualitatively similar results (Fig. S11).”

We also added a new figure to the SI, presenting the results of simulations with 4 interactions per generation (Fig S11 in the SI).

From my side, I find the assumptions about microbe transmission rather peculiar. The hosts cannot clear the microbe and vertical transmission is perfect, so there are no uninfected hosts. Yet, once microbe alpha is transmitted to the host carrying microbe beta or vice versa, it results in elimination of the other strain, so no persistent multiple infection is possible. So, in effect, the host cannot get rid of neither microbe alpha nor beta, but microbe alpha infecting host already infected with beta eliminates the latter and vice versa (with some probability < 1). At least a preliminary investigation of relaxing these assumptions – allowing for vertical transmission with probability < 1 and/or for simultaneous infection with the two microbes should be carried out. The latter case might in particular be interesting, because it would generate situations where microbe alpha induces an altruistic act, but it happens to be microbe beta that is transmitted and thus profits from the altruism. Without considering such broader – and possibly biologically more realistic – assumptions, the statements that the model explains cooperation “in a wide variety of organisms” (l. 34-35 or 303-304) is simply not justified. It may be true that this holds under a broad range of parameters, but in the framework of a highly specific model.

We first note that in the original ms there were two ways to get rid of a microbe. First, by getting infected by another host. And in this sense there is no inherent advantage to microbe α (cooperation-inducing microbe) as we focus on the case of horizontal transmission advantage to microbe β . The second mechanism is mutations. In our models we include mutations both in the alleles and in the

microbes, which are another way to get rid of a specific microbe (in this paper we show results for $N \cdot \mu = 0.1, 1, 10$).

Nevertheless, we agree with the editor that it would be more realistic to account for imperfect vertical transmission, and we now include this in our model. The results show that cooperation can evolve even when considering imperfect vertical transmission.

We discuss this in the results (L354):

“We further examined the effect of imperfect vertical transmission, modeled similarly to ([29] and its SI): with probability ρ the offspring inherits its parent microbes, and with probability $1 - \rho$ it inherits microbes from a random host from the parent generation (see further details in SI4). We found that cooperation can evolve and be maintained even when the vertical transmission is far from perfect (Fig. S10).”

And in the revised SI:

(L318): “**Imperfect vertical transmission.** We modeled imperfect vertical transmission similarly to [Lewin-Epstein et. al. 2017; see also SI]: during reproduction, with probability ρ , the offspring inherits its parent microbes, and with probability $1 - \rho$ it inherits microbes from a random host from the parent neighborhood (in a fully-mixed population the parent neighborhood includes the entire parent population).”

We also added a figure showing simulation results with imperfect vertical transmission (figure S10 in the SI).

As for coinfection, we agree that coinfection can yield even more complex dynamics. However, note that the non-manipulating counterpart would not be a classic “cheat” here: since it is the manipulated *host* that pays the cost of cooperation, it would apply to all the microbes it carries. Coinfection with a non-manipulating microbe would result in decreased transmission probability of microbe α to a host carrying microbe β – but we already focus on the case that microbe α has a transmission disadvantage. Being coinfecting with a non-manipulative microbe is likely to increase the effective disadvantage, but not change the dynamics qualitatively.

We added a reference to this possible extension in the discussion (L371):

“In natural populations, hosts’ behavior might be affected by several loci in the genome, and by the composition of the microbial community, including the cases of co-infection with multiple microbes with different effects.”

We also revised the mentioned sentences (l. 34-35 or 303-304):

(L34): “Our results suggest a mechanism for the evolution and maintenance of cooperation that may be relevant to a wide variety of organisms, including cases that are difficult to explain by current theories.”

(L366): “We find that in finite-size populations, cooperation can also be maintained in polymorphism, in the presence of either spatial structure or mutations of the microbes and the hosts.”

Reviewers 1 and 2 also see potential to improve the presentation. Notably, the model results could be better intuitively explained and dissected. In addition to the specific example mentioned in the reviews, it is not clear to me to what degree the more favorable conditions for altruism in spatially viscous population in the simulations are due to kin selection in the host versus the microbe. One way of resolving it would be to run some simulations in which either the host or the microbe genotypes are spatially randomized every generation. The reviewers point to several other potentially improvements and issues that the authors should carefully consider.

We thank the editor for this comment. We added two paragraphs to the results section, and a new figure (S5) showing the effect of microbe shuffling at different stages in the life cycle (note that host shuffling is already included in the “fully mixed” model/simulations).

(L190):

“Two major factors support the evolution of microbe-induced cooperation. First, the link between interaction and horizontal transmission, which enables the microbes to direct some of their host resources towards another host, that could be inhabited by their transmitted kin. Second, the ability of the microbes to transmit both horizontally and vertically: while horizontal transmission allows the microbes to help their future kin, vertical transmission allows the microbes to enjoy the increased fitness of a host that received help.”

(L239):

“Cooperation in our model can evolve even under transmission disadvantage. In that case cooperation cannot evolve by infectivity alone. Kin selection among hosts is not a major factor either, as we here

consider fully-mixed populations. It is rather kin selection at the microbes' level that enables the evolution of cooperation: microbe-induced cooperation evolves due to the ability to preferentially direct the cooperation benefit towards other (future) cooperators, *even in fully-mixed populations.*"

We now also refer to this remark in the stochastic results when comparing fully-mixed with spatially structured populations (L315):

"We find that both mutations and spatial structure dramatically widen the range of parameters that allow the maintenance of cooperation in polymorphism, when $\delta < c$ and (1) is satisfied. Most of the effect is because both mechanisms reduce the probability of stochastic extinction of genotypes (compare area III of Fig. 4b and 4c to 4a; see also supplementary video file, displaying the dynamics of one spatially-structured population). However, even in populations with mutations spatial structure yields slightly higher proportions of cooperation, probably due to the additional effect of kin selection among hosts (compare area III of Fig. 4d and 4c; see further analysis of the effects of host and microbe kin-selection in Fig. S5)."

In the new figure S5 – we explain the effects of kin selection on the microbes' and hosts' level by comparing spatially structured population with and without shuffling of microbes (shuffling of hosts is already included in the "fully mixed" populations, Figs 2,4).

Additional comments:

The ability to manipulate host behavior is clearly not limited to microbes. Nonmicrobial parasites, commensals or symbionts can also be transmitted horizontally, what is said in lines 76-83 clearly applies to many multicellular parasites. OK, I see the authors say in passing in the Discussion that their results apply to those other taxa, but why not to state so from the beginning rather than insisting first that microbes are somehow different?

We thank the editor for this comment, and we agree that it is best to point this at the introduction rather than in the discussion. We now mention this in the introduction (L74):

"Note that in this work we use the term microbe in the most general sense, referring to an element that inhabits an organism, can affect its behavior, and can be transmitted both vertically and horizontally.

Our results can be relevant to any element that applies to these characteristics (e.g. plasmids, viruses, multicellular symbionts, etc.).”

I. 320-322: This does not seem to be an example of microbes manipulating their multicellular hosts, but of a horizontally transferable genetic element affecting the cooperative behavior of their microbial carriers. So it does not fit the preceding sequence. On the other hand it seems to me that it does fit the general logic of the model. It may actually be the most biologically plausible case where the basic premise of this model applies (although the specific assumptions about transmission might still not be plausible). So this is another reason to specify the idea in more general terms of horizontally and vertically transmitted infectious agents.

We revised this paragraph (L382):

“There is already some evidence suggesting that microbes can affect cooperative behavior in their host. Among eukaryotic hosts, *Lactobacillus* is a promising candidate: accumulating evidence demonstrates that *Lactobacillus* decreases stress-induced anxiety-like behavior, potentially increasing the tendency of its host to interact with other individuals, and promoting cooperative behavior [82-85]. In plants, carbon sharing among trees is, at least partially, mediated by mycorrhizal fungi that form networks connecting neighboring tree roots [86, 87]. This behavior can be considered as cooperation among trees, induced by their fungi. And if considering bacteria as hosts that inhabit mobile genetic elements, it was shown that genes responsible for public good secretion are prevalent on mobile elements that can be transmitted both horizontally and vertically between host cells [68, 69]. Our results suggest that such effects may be common in many other systems as well, and call for experimental tests.”

I. 180-188: an important piece of information missing in that otherwise clear intuitive explanation is that the polymorphism in the host is maintained by a balance between its disadvantage when infected with alpha and its advantage when infected with beta. This directly explains the equilibrium property in I. 179 – at this frequency of alpha infection the fitness of R and S hosts is equal.

We thank the editor for this comment. We added this explanation to the mentioned paragraph (L211):

“The polymorphism in the host alleles is maintained by a balance between the disadvantage of resistance (paid by hosts carrying allele *R*) and the disadvantage of cooperation (paid by those *S* hosts that carry microbe α).”

I. 323: the authors should follow the rules of biological nomenclature like capitalizing genus names
Fixed

Reviewer(s)' Comments to Author:

Referee: 1

Comments to the Author(s)

The authors investigate the evolution of a microbe that can induce cooperation in its hosts and the evolution of host resistance/susceptibility to microbial influences. They find cases where cooperation is present in some stable, positive fraction of the population as well as cases where the levels of cooperation show cyclical dynamics with other combinations of host and microbial genotypes. This builds on previous work investigating the evolution of microbe-induced cooperation where hosts were unable to resist their microbes effects. Adding hosts resistance adds an important layer of biological realism and produces interesting new dynamics. The question of when hosts would allow themselves to be “coerced” into cooperating is also an extremely interesting one. The particular model the authors use is a good way to investigate the question. There are some minor things that could be changed, mainly to improve clarity.

I think the most important issue for clarity has to do with the order of events when hosts interact. In the model, horizontal transmission occurs when hosts meet to play a prisoner's dilemma game. It is not clear from the main text whether transmission occurs before or after play, although equations 1 – 4 in the supplement suggest that transmission occurs after play (or, if it occurs before play, that any newly-acquired microbe does not affect the host's behavior). I think this needs to be made explicit in the main text to avoid confusion with another possible model of how relatedness between microbes could allow the spread of cooperation – if microbes were to transmit and influence their new host's behavior prior to play, cooperators could potentially increase their chances of interacting with another cooperator. We thank the reviewer for this comment and agree that this point was not clear enough. We now emphasize this by adding the following part to the model:

(L142): “We assume that during an interaction, a host behaves according to the allele-microbe combination it carried before the interaction. If horizontal transmission occurs, the new microbes establish and start affecting host behavior right after the current interaction.”

Results:

Figure 1 is very clear and helpful for understanding the model.

Thanks!

Figure 2: Would it be possible to have a supplementary figure showing the expected fractions of other host-microbe combinations? It would be interested to see how these change over parameter space as well.

We thank the reviewer for this suggestion. We added to the supplementary information a new figure (Fig. S3) showing all four host type frequencies, with same model parameters as in Fig. 2b of the main text.

Figure 3: These plots are extremely helpful for understanding the dynamics of the system. Having the start point highlighted is really useful for visualizing the direction of evolution. Would it be possible to highlight the location of the equilibrium as well?

Thanks! The revised figure includes markings of the equilibria (blue dots in Fig. 3b,d).

Figure 4: I am intrigued by the shape of the region where cooperation does not go extinct in a and b. Is there any intuition about why it looks like this?

Indeed, this is an interesting point to which we now specifically refer in the results section (L343):

“We find that without mutations, the proportion of cooperators is not monotonic in δ (see area III in Fig. 4a,b). When δ increases towards c , αR takes over the population frequently, while when δ decreases towards 0, fixation of βS becomes common (see SI2 and Fig. S9). When allowing mutations, this pattern vanishes (compare Fig. 4a to 4c, and 4b to 4d).”

We also added a new figure to the SI (Fig. S9), presenting the dynamics that yield this shape.

Discussion:

Lines 330 – 338: I am excited about the possibility of this model being applicable to gene-culture

coevolution. The way cultural transmission is described here, I am not sure if it exactly fits the model. If individuals must see their interaction partner exhibit a behavior to acquire it horizontally, then “ α R” individuals seem like they will transmit the “ β ” cultural trait, rather than “ α .”

We thank the reviewer for this remark. We agree that cultural settings would not necessarily be the same as ours. We thus added the following comment (L400):

“Differently from our model, a resistant genotype would not express cooperative behavior and thus would probably not transfer it through imitation.”

Methods:

Supplementary equations 1 – 4 provide a lot of insight into the model, and it would be great if they or their simplified versions were in the methods section.

Due to lack of space, we had to leave the equations and the methods in the SI.

What fractions of host types were the simulations initialized with?

We thank the reviewer for pointing out that this detail was missing. We added to the caption of figure 4 and to the relevant SI figures the following sentence:

“All simulations were initialized with equal proportions of the four host types.”

I imagine the oscillations in the fraction of infected hosts are not so regular that this is a problem, but does the proportion of cooperators at the end of 100 simulations cover the range of variation in the fraction of cooperators?

We thank the reviewer for this comment, and we added clarification of these results. We added figure S7 (in the SI) showing, for each δ and b/c values, what proportion of the simulations ended in polymorphic result (as oppose to fixation/extinction).

We added a reference to this figure in the caption of figure 4 (L338):

“The presented intermediate averages in area III usually represent population polymorphisms, apart from panel (a) (Fig. S7).”

Lines 374-377: I don't understand how interactions and transmission work in the simulations with spatial structure. What is the impact of choosing an individual to initiate an interaction? What is the fitness of an individual who, by chance, never interacts with anyone? How do transmission and microbe effects work when there are potentially multiple interactions within a generation? For example, can an α S host

interact with a βS neighbor, become infected, and then, in its interaction with a second neighbor, re-acquire an α microbe? Does this host behave non-cooperatively in its interaction with the second neighbor?

We thank the reviewer of pointing that this was not clear enough. We revised the presentation of the simulation workflow – both fully mixed and spatially structured: (SI4 L247-L316).

Supplement:

I think the authors' decision to show only the extended model is very reasonable. For readers looking for the simplified model, it would be good to state that only the extended model will be discussed in S1 and S2 and explain how readers can get the results for the simplified model from the extended.

We added the following sentence to the introduction of the SI1 (SI L18):

“Note that in the main text we show results for the case of $\delta_\alpha = \delta_\beta = \delta$, and analysis of the equilibrium in that case is presented in section 3.2 of this SI.”

The explanations of how the Jacobians were found are excellent.

It would be great if the descriptions of the eigenvalues for invasion gave some explanation about the role of these values in the analysis and what each corresponds to.

We now elaborate on each eigen value, stating that if it is greater than 1, then invasion will occur.

For example (SI L53):

“And the eigen values are $\frac{1+b-\delta_\alpha}{1+b-c}$, $\frac{(1-T_\alpha)(1+b)+T_\beta(1-c)}{1+b-c}$ and $\frac{(1-T_\alpha)(1+b-\delta_\beta)}{1+b-c}$.

Thus, when αS hosts are at fixation:

- If $\frac{1+b-\delta_\alpha}{1+b-c} > 1$ (namely, $\delta_\alpha < c$) then αR hosts can invade the population
- If $\frac{(1-T_\alpha)(1+b)+T_\beta(1-c)}{1+b-c} > 1$ (namely, $\frac{b}{c} < \frac{1-T_\beta}{T_\alpha} + \frac{T_\beta-T_\alpha}{T_\alpha c}$) then βS hosts can invade the population
- If $\frac{(1-T_\alpha)(1+b-\delta_\beta)}{1+b-c} > 1$ then βR hosts can invade the population”

Lines 78-80: This sentence is confusing. Some explanation of how the intersection of conditions was found that allows the coexistence of multiple types would be wonderful.

We thank the reviewer for this comment. We now elaborate on the intersection of the conditions, and the characteristics of the subset of conditions we were looking for (SI L97):

“Altogether, we obtained 12 conditions for invasion of each host type, to a population dominated by another host type.

- We first note that fixation of βS is stable unless: $\delta_\beta < 0$, $\frac{b}{c} > \frac{1-T_\beta}{T_\alpha} + \frac{T_\beta-T_\alpha}{T_\alpha c}$, or $(1-T_\beta)(1-\delta_\alpha) > 1$. Since we focus on $0 < T_\beta, \delta_\alpha < 1$, only the second condition can disrupt the stability of βS fixation. Thus fixation of βS is stable unless $\frac{b}{c} > \frac{1-T_\beta}{T_\alpha} + \frac{T_\beta-T_\alpha}{T_\alpha c}$ (condition 1 of the main text). In addition we note that βS hosts can invade a βR -population as long as $\delta_\beta > 0$.
- Second, we note that $\frac{b}{c} > \frac{1-T_\beta}{T_\alpha} + \frac{T_\beta-T_\alpha}{T_\alpha c}$ and $\frac{(1-T_\alpha)(1+b-\delta_\beta)}{1+b-c} > 1$ contradict each other. Note that the first term can be rewritten as $T_\alpha b > c(1-T_\beta) + (T_\beta - T_\alpha)$ while the second term can be rewritten as $T_\alpha b < [c(1-T_\beta) + (T_\beta - T_\alpha)] - [T_\beta(1-c) + \delta_\beta(1-T_\alpha)]$. Thus, if αS hosts can invade βS population, then βR hosts cannot invade αS population.
- Lastly, we note that if $\frac{(1-T_\beta)(1-c)}{1-\delta_\beta} > 1$, then c must be smaller than δ_β . If in addition $\delta_\alpha < c$, then altogether we get that $\delta_\alpha < c < \delta_\beta$ – a scenario which we neglect, as we focus on cases where the resistance cost in the presence of manipulating microbe is equal or higher to the resistance cost in the absence of this microbe.”

S3.2 explains the polymorphic equilibria and the dynamics of the system really well.

I am curious about the conditions where the fractions of hosts of each type oscillate vs. reach equilibrium. Would it be possible to discuss which initial values lead to each outcome? Or perhaps to show a sample plot highlighting the initial fractions of each host type that lead to oscillation vs. equilibrium?

This is indeed interesting, but unfortunately we were not able to derive analytic conditions that determine the dynamics. We thus added a new figure to the SI (Fig, S4), showing which initial population compositions leads to convergence, and which to divergence.

The supplementary video was extremely helpful for visualizing the dynamics of the system. I couldn't find a reference to it in the main text, and I think one would be great.

There was a reference to the supplementary video in L326 in the main text, but we now added a short description of what this file contains.

Referee: 2

Comments to the Author(s)

This paper presents a model of cooperation (altruism in the inclusive fitness sense, modelled as a 2-player single-shot Prisoner's Dilemma) is induced in hosts through transmitted microbes. This builds on a model presented in a prior paper published by the authors: Lewin-Epstein O., Aharonov R., Hadany L. 2017 Microbes can help explain the evolution of host altruism. Nature communications 8, 14040 . That paper shows conditions under which a cooperative trait induced by horizontally transmitted microbes can spread, even when the hosts themselves are not related (in effect both because of horizontally transmission through infection and the establishment of relatedness at the locus for cooperation carried by the microbes).

The advance in the present paper is to now consider that hosts may be able to carry a gene that confers resistance to the effects of the microbes. The results are that this leads to rock-paper-scissors dynamics in terms of invasability. Even when the cost of resistance is less than the cost of the induced cooperation in the Prisoner's Dilemma, a polymorphic equilibrium level of cooperation can result.

The results are potentially interesting theoretically and empirically. However, their presentation and interpretation needs to be strengthened:

1. Theoretically, the selective forces favouring cooperation in the model are not well explained. What exactly is the role of relatedness, both between microbes, and between hosts? What is the role of infectivity (the transmission rates)? Can you disentangle these?

We thank the reviewer for this comment. We revised the results section to include an explanation for the mechanism that enables the evolution of microbe-induced cooperation, and added a new figure (Fig. S5).

(L190):

“Two major factors support the evolution of microbe-induced cooperation. First, the link between interaction and horizontal transmission, which enables the microbes to direct some of their host

resources towards another host, that could be inhabited by their transmitted kin. Second, the ability of the microbes to transmit both horizontally and vertically: while horizontal transmission allows the microbes to help their future kin, vertical transmission allows the microbes to enjoy the increased fitness of a host that received help.“

(L239):

“Cooperation in our model can evolve even under transmission disadvantage. In that case cooperation cannot evolve by infectivity alone. Kin selection among hosts is not a major factor either, as we here consider fully-mixed populations. It is rather kin selection at the microbes’ level that enables the evolution of cooperation: microbe-induced cooperation evolves due to the ability to preferentially direct the cooperation benefit towards other (future) cooperators, *even in fully-mixed populations.*“

We now also refer to this remark in the stochastic results when comparing fully-mixed with spatially structured populations (L315):

“We find that both mutations and spatial structure dramatically widen the range of parameters that allow the maintenance of cooperation in polymorphism, when $\delta < c$ and (1) is satisfied. Most of the effect is because both mechanisms reduce the probability of stochastic extinction of genotypes (compare area III of Fig. 4b and 4c to 4a; see also supplementary video file, displaying the dynamics of one spatially-structured population). However, even in populations with mutations spatial structure yields slightly higher proportions of cooperation, probably due to the additional effect of kin selection among hosts (compare area III of Fig. 4d and 4c; see further analysis of the effects of host and microbe kin-selection in Fig. S5).“

In the new Fig. S5 we explain the effects of kin selection on the microbes’ and hosts’ level by comparing spatially structured population with different forms of shuffling of microbes (shuffling of hosts is already included in the “fully mixed” populations, Figs 2,4).

2. The paper needs to connect more with the literature on horizontal transmission of cooperative traits through plasmids. In particular, the paper “Mc Ginty SE’, Lehmann L, Brown SP, Rankin DJ. 2013 The interplay between relatedness and horizontal gene transfer drives the evolution of plasmid-carried public goods. Proc R Soc B 280: 20130400” would help to explain the effects of relatedness and

infectivity. There are also many other papers in this area, and these should be discussed in the introduction of the paper.

We thank the reviewer for this comment. We revised the introduction by adding a paragraph, presenting previous works on mobile genetic elements, and emphasizing the novelty in our work (L96):

“Previous studies examined the evolution of public goods genes that are encoded on mobile genetic elements that can be transmitted vertically and horizontally between bacteria ([67-70]). These studies explored the role of transmission and relatedness on the evolution of bacterial cooperation, both empirically and theoretically. Our work is different in several ways. First, we broaden the perspective and claim that in fact, the evolution of cooperation by horizontally-transmitted elements is relevant to almost any species, via its microbiome, or any other symbiont. Second, we emphasize the link between social interaction and horizontal transmission, which we show, enables the evolution of cooperation under wide conditions. Last, we study host-microbe coevolution with respect to cooperation, by accounting for both microbial and host alleles.”

3. The model relies on a dyadic interaction, where crucially horizontal transfer occurs immediately after the Prisoner’s Dilemma game, between the same pair of individuals. I suspect that this strongly favours the spread of cooperation compared to the case where transmission is not directly coupled to the Prisoner’s Dilemma game. How realistic empirically is the scenario that microbial exchange occurs during the Prisoner’s Dilemma interaction?

In general, whenever there is an interaction that involves close contact, microbes can be transmitted. Cooperating interactions in many cases indeed involve close interaction between individuals and this proximity allows horizontal transmission: direct feeding, sharing food source, grooming, allowing another individual to co-shelter etc.

We now stress this point in the introduction (L86):

“Yet, as opposed to genes of multicellular organisms, microbes can also be transmitted horizontally during interactions between hosts [60-63]. In fact, interactions among hosts, such as feeding, grooming, sharing resources and co-sheltering, involve close proximity between the hosts, and thus serve as a platform for microbial transmission [64-66].”

4. What are the effects in the model of relaxing the assumption in 3. ?

The link between behavioral interaction and horizontal transmission is the major force in our model. Thus, if breaking this link completely – microbe-induced cooperation cannot evolve. In fully-mixed populations, if each hosts pairs randomly with another host with whom it plays the prisoners' dilemma, and then it pairs with a third random host with whom it interacts in a way that allows only horizontal transmission – microbe-induced cooperation indeed cannot evolve (new Fig. S5d). But if applying the same settings in the spatially-structured model, where both interaction and transmission occur between neighbors, microbe-induced cooperation can evolve, but the b/c threshold increases significantly. In this case hosts interact only with their neighbors and thus there is still a reasonable probability to perform both interactions (prisoners' dilemma and horizontal transmission) with the same partner (probability $\geq 1/8$). We now show this in Fig. S5c.

5. Many interactions empirically involve n individuals rather than dyads. Would the mechanism of microbial transmission still be expected to work in that case?

We thank the reviewer for this question which we did not address at first. In order to address this point within the scope of our model we now include analysis of simulations with multiple interactions within each generation, and a new figure (Fig. S11).

Our stochastic simulations showed robustness of the model to multiple interactions. Having multiple interactions increases the effect of horizontal transmission. When considering transmission disadvantage of microbe α , then as there are more interactions per generation, the transmission advantage of β plays a larger role, and cooperation can evolve under narrower range of parameters. Nevertheless, we found that when modeling 4 interactions per generation, if setting $T_\alpha = T_\beta \cdot 0.9^{(1/4)}$ and $c = 0.05/4$, we obtained very similar results as when simulating one interaction per generation with $T_\alpha = T_\beta \cdot 0.9$ and $c = 0.05$.

We now mention this in the results section (L361):

“Finally, we extended the model to account for multiple interactions (K) per host per generation and found qualitatively similar results (Fig. S11).“

and also in the simulation workflow description in SI4.1 (L249).

We also added a new figure presenting simulation results with four interactions per host per generation – Fig. S11 in the SI.

6. Are there any concrete empirical examples of microbes inducing cooperation in their hosts?

We do not know of any direct evidence. We hope to bring this theory to the attention of experimental biologist and to encourage the empiric study of microbe-induced cooperation.

Yet, we have encountered some examples which we believe can be interpreted as cooperation-inducing microbes, at least to some degree. And we now emphasize them in the discussion (L382):

“There is already some evidence suggesting that microbes can affect cooperative behavior in their host. Among eukaryotic hosts, *Lactobacillus* is a promising candidate: accumulating evidence demonstrates that *Lactobacillus* decreases stress-induced anxiety-like behavior, potentially increasing the tendency of its host to interact with other individuals, and promoting cooperative behavior [82-85]. In plants, carbon sharing among trees is, at least partially, mediated by mycorrhizal fungi that form networks connecting neighboring tree roots [86, 87]. This behavior can be considered as cooperation among trees, induced by their fungi. And if considering bacteria as hosts that inhabit mobile genetic elements, it was shown that genes responsible for public good secretion are prevalent on mobile elements that can be transmitted both horizontally and vertically between host cells [68, 69]. Our results suggest that such effects may be common in many other systems as well, and call for experimental tests.”

7. The paper ends by stating that it makes precise testable predictions. The discussion should be more explicit about what exactly these are.

We thank the reviewer for this comment, and we now better phrase this part (L407):

“These results strengthen the theory of microbe-induced cooperation, and may help explain occurrences of cooperation that are difficult to explain by current theories. Our results provide verifiable predictions that can be tested in future experimental efforts: first, that altering the composition of microbial communities (e.g. by antibiotics [93-95], probiotics [96, 97], pesticides [98] and herbicides [99], etc.) may affect the hosts’ social behavior; and second, that polymorphism with respect to cooperative behavior can be expected in natural populations, originating in a conflict between host genes and microbes.”

Referee: 3

Comments to the Author(s)

The manuscript investigates the interesting idea that microbes could manipulate the cooperation of their hosts with other hosts. The authors take a coevolutionary approach to understand the stability of microbe-mediated host cooperation by including hosts that are susceptible and resistant to manipulation and microbes that do and do not manipulate hosts. The authors find that it is possible for microbe-mediated cooperation to persist in populations across a wide range of conditions. They also find that the inclusion of mutation and spatial structure increase the parameter space in which cooperation persists.

These findings are useful because they could explain the presence of cooperation in cases where it may not otherwise have been expected based on current theory. The manuscript is novel in this way. The manuscript will be of interest to those researching cooperation, social behaviour in general, evolution and microbe-host interactions. It is also well-written and the logic is clear. I also appreciate the discussion towards the end, acknowledging what the models do not incorporate, e.g. the composition of the microbial community in the host.

It will be interesting to consider how co-infection of a manipulating microbe with its non-manipulating counterpart will affect dynamics. Presumably, if this non-manipulating microbe can benefit from the cooperation induced by the manipulating microbe, it will act as a 'cheat', which could affect the conditions under which cooperation will persist.

We agree with the reviewer that coinfection can yield even more complex dynamics. However, note that the non-manipulating counterpart would not be a classic "cheat" here: since it is the manipulated *host* that pays the cost of cooperation, it would apply to all the microbes it carries. Coinfection with a non-manipulating microbe would result in decreased transmission probability of microbe alpha to a host carrying microbe beta – but we already focus on the case that microbe alpha has a transmission disadvantage. Being coinfecting with a non-manipulative microbe is likely to increase the effective disadvantage, but not change the dynamics qualitatively.

We thus added this comment into the discussion, where we present possible future extensions to our model (L371):

"In natural populations, hosts' behavior might be affected by several loci in the genome, and by the composition of the microbial community, including the cases of co-infection with multiple microbes with different effects."

specific points:

Line 39: As far as I understand currently-accepted terminology, 'cooperation' can include interactions where there is no cost, while your definition and model explicitly state a >0 cost. This is defined as altruism. You might want to mention this.

We agree that both terms are used interchangeably (even by Hamilton himself). We prefer using the term cooperation, but we added a reference to the alternative terminology (L66):

“We use the term “cooperation” throughout, but note that similar phenomena were referred also as “altruism” in the literature.”

Line 278: "maintaining" should be "maintain".

Fixed, thanks!

Line 383, "where" should be "were".

Fixed, thanks!

Appendix B

Dear Editor,

We would like to thank again the associate editor and the reviewers for their comments.

Following the questions and suggestions, we made the following change:

- We performed an analysis of spatially-structured simulations with multiple interactions per host per generation, where each host's fitness was normalized by the number of interactions in which it participated. The results are presented in a newly added figure (figure S12). We found very similar results to those of populations with one interaction per host per generation (figure 4) and to those of populations with multiple interactions per host per generation with a different normalization of the fitness (figure S11).

Below are the comments, each followed by our reply and the details of the revisions made to address the comment.

We believe we have addressed the editor's and the reviewers' comments and suggestions and we hope you find our manuscript suitable for publication in Proceedings of the Royal Society B.

Sincerely,

Lilach Hadany

Associate Editor Board Member

Comments to Author:

The reviewers are satisfied with the revisions done by the authors. However, reviewer 1 still raises one issue that the authors should address in discussion and possibly be making some exploratory simulations.

Reviewer(s)' Comments to Author:

Referee: 1

Comments to the Author(s).

I really like the authors' changes, which I think have made things a lot clearer. In particular I think the new model explanation is really clear and gives a very complete description of the model.

I do have a small comment about the effect of multiple interactions on fitness in the simulations, mainly the ones of spatially structured populations. Because the costs and benefits of multiple interactions aren't normalized by the number of interactions, the expected fitness of an individual interacting with the same distribution of partners can change depending on the number of partners it has. The authors recognize this and deal with it by dividing the cost of cooperation by the expected number of interactions in Figure S11. This is great for the well-mixed simulations (where number of interactions is fixed). It's probably not a problem for the spatially structured simulations either, given how robust the simulations are to other changes in the model. However, the relationship between expected fitness and number of interactions might have an impact on hosts in these simulations, since hosts have a variable number of interactions. I think it's worth mentioning this in the text or possibly doing a quick check of what happens when the expected payoff isn't a function of the number of interactions, if that's not too much trouble.

We thank the reviewer for this comment. We ran simulations of spatially-structured populations where the cost of cooperation remains $c = 0.05$ (and not $0.05/4$ as in figure S11), and the fitness of each individual is normalized by the number of interactions it participated in. The results, which indeed turned quite similar to those presented in figure S11, are presented in the newly added figure S12.

A few other very minor things:

Line 115: I think "to the microbial effect" or similar probably needs to remain to clarify that "resistance" doesn't refer to resistance to infection.

Fixed

Line 430: “considering bacteria as hosts that inhabit mobile genetic elements” should be “considering bacteria as hosts inhabited by mobile genetic elements”

Fixed

General: “their host behavior” should be “their host's behavior”

Fixed

Referee: 2

Comments to the Author(s).

The authors have modified the paper to address my concerns.